# Delineating the Dynamic Transcriptome Response of mRNA and microRNA during Zebrafish Heart Regeneration

**DOI:** 10.3390/biom9010011

**Published:** 2018-12-28

**Authors:** Hagen Klett, Lonny Jürgensen, Patrick Most, Martin Busch, Fabian Günther, Gergana Dobreva, Florian Leuschner, David Hassel, Hauke Busch, Melanie Boerries

**Affiliations:** 1Institute of Molecular Medicine and Cell Research, University of Freiburg, 79104 Freiburg, Germany; hagen.klett@gmail.com; 2Department of Cardiology, Angiology and Pneumology, University Hospital Heidelberg, 69120 Heidelberg, Germany; Lonny.Juergensen@med.uni-heidelberg.de (L.J.); Patrick.Most@med.uni-heidelberg.de (P.M.); Martin.Busch@med.uni-heidelberg.de (M.B.); f.guenther@rbk.de (F.G.); Florian.Leuschner@med.uni-heidelberg.de (F.L.); david.hassel@gmail.com (D.H.); 3German Centre for Cardiovascular Research (DZHK), Partner Site Heidelberg/Mannheim, 69120 Heidelberg, Germany; 4Center for Translational Medicine, Jefferson University, Philadelphia, PA 19107, USA; 5Department of Anatomy and Developmental Biology, Medical Faculty Mannheim, Heidelberg University, 68167 Mannheim, Germany; gergana.dobreva@medma.uni-heidelberg.de; 6Luebeck Institute of Experimental Dermatology and Institute of Cardiogenetics, University of Luebeck, 23562 Luebeck, Germany; hauke.busch@uni-luebeck.de; 7Comprehensive Cancer Center Freiburg (CCCF), University Medical Center, Faculty of Medicine, University of Freiburg, 79106 Freiburg, Germany; 8German Cancer Consortium (DKTK), Partner Site Freiburg, 79104 Freiburg and German Cancer Research Center (DKFZ), 69120 Heidelberg, Germany

**Keywords:** heart regeneration, zebrafish, cryoinjury, dynamic transcriptome, miRNA

## Abstract

Heart diseases are the leading cause of death for the vast majority of people around the world, which is often due to the limited capability of human cardiac regeneration. In contrast, zebrafish have the capacity to fully regenerate their hearts after cardiac injury. Understanding and activating these mechanisms would improve health in patients suffering from long-term consequences of ischemia. Therefore, we monitored the dynamic transcriptome response of both mRNA and microRNA in zebrafish at 1–160 days post cryoinjury (dpi). Using a control model of sham-operated and healthy fish, we extracted the regeneration specific response and further delineated the spatio-temporal organization of regeneration processes such as cell cycle and heart function. In addition, we identified novel (miR-148/152, miR-218b and miR-19) and previously known microRNAs among the top regulators of heart regeneration by using theoretically predicted target sites and correlation of expression profiles from both mRNA and microRNA. In a cross-species effort, we validated our findings in the dynamic process of rat myoblasts differentiating into cardiomyocytes-like cells (H9c2 cell line). Concluding, we elucidated different phases of transcriptomic responses during zebrafish heart regeneration. Furthermore, microRNAs showed to be important regulators in cardiomyocyte proliferation over time.

## 1. Introduction

Cardiovascular diseases claim more lives than all forms of cancer combined with one reason being the limited capability of heart regeneration following cardiac damage [1]. In contrast, neonatal mice are able to regenerate their hearts after LAD-ligation [2,3] and zebrafish can do so after resection [4,5] and after induced necrosis through cryoinjury [6,7]. The massive cell death and scar formation induced by cryoinjury results in long term functional recovery [8,9], making it particularly useful to simulate myocardial infarction and study regeneration processes. Yet, zebrafish have to undergo surgery, which impedes the study of heart regeneration processes, and sham-operated fish are encouraged as controls. Delineating the transcriptome response in zebrafish during cardiac regeneration elucidates important regulatory mechanisms. These mechanisms can potentially become activated in humans and contribute to a decreasing risk of recurrent cardiac diseases, such as myocardial infarction, and improve life post heart injury dramatically.

In zebrafish, proliferating and dedifferentiating cardiomyocytes (CM) invading the area of injury are considered to be the major source for heart regeneration [10,11]. There is evidence that cell cycle processes are activated after myocardial injury [12]. In contrast, this ability is normally blocked or cannot be executed completely for mammalian hearts [13]. Yet, genetic manipulation experiments of cell cycle genes or of proliferation regulators suggest that this ability can be reactivated [13,14]. These findings encourage the search for transcriptomic regulators pushing CM to re-enter cell cycle processes in non-regenerating species. In search of underlying mechanisms, studies have identified Notch [5], BMP [15], and PDGF [16] signaling as essential pathways for cardiac regeneration in zebrafish. Among cell cycle regulators specifically, there are transcription factors, such as tbx5, gata4 [11] and hand2 [17] as well as microRNAs (miRNAs) [18,19] that were shown to play an important role in the post-injury cardiovascular response.

MiRNAs are abundant, cell specific ~18–22 nucleotide non-coding RNAs that regulate gene expression at a post-transcriptional level. Depending on their “seed” sequence they can pair with matching mRNAs, which results in transcript destabilization, translational repression or both. Yet, “seed” pairing alone is not reliable for miRNA-target predictions [20] but can be enhanced by integrating expression correlations between mRNA and miRNAs [21]. Moreover, many miRNAs can target one mRNA and multiple mRNAs can be targeted by one miRNA, making functional interpretation delicate [19]. However, their conservation in nucleotide sequence across species and also their target genes across mammals make them interesting players in translational research [22]. In zebrafish, the miR-133 family and miR-101a can regulate heart regeneration by targeting cell cycle factors. Depletion of these miRNAs results in enhanced regeneration while over-expression impedes regeneration processes [23,24]. Comparative transcriptome profiling of injured zebrafish and mouse hearts further identified miR-26a as negative regulator of CM proliferation [25], highlighting their potential in targeted therapy after myocardial infarction.

We monitored the mRNA and miRNA response after cryoinjury (1–160 dpi) to decipher the genetic mechanisms and gain insight in the complex interplay between miRNAs and mRNAs during zebrafish heart regeneration. A time-dependent linear model was used to extract cryoinjury specific changes from surgical and age-related background changes. This enabled us to identify the spatio-temporal organization of the transcriptome. We identified the most important dynamic miRNA regulators for zebrafish heart regeneration through the prediction of miRNA-mRNA interactions. Lastly, in a cross-species effort, we validated the dynamics of the target genes of the most important miRNAs in differentiating rat myoblasts (H9c2 cell line). Our findings confirm previous results from literature and present novel key miRNA regulators with a potential translational impact.

## 2. Results

### 2.1. Extracting Cardiac Regenerative Transcriptome Responses after Cryoinjury in Zebrafish

To study the complex interplay between miRNAs and their target genes during cardiac regeneration in zebrafish, we induced myocardial necrosis in adult zebrafish by application of a cryoinjury model to simulate myocardial infarction [7,8]. A schematic workflow of the study design and the necessary analytic steps are given in Figure 1.

The dynamic response was captured at 1, 4, 7, 14, 21, 30, 45, 60, 120 and 160 days post injury (dpi). The fish were sacrificed between 220 and 463 days of age (DOA) corresponding to 17% and 36% of their expected average lifespan of 3.5 years [26]. Therefore, we added four groups of healthy fish at 154, 302, 378 and 463 DOA to control for potential age-related transcriptomic changes (Appendix A). Finally, we acquired one group of sham-operated fish at 1 day post-surgery to control for transcriptomic changes induced by the surgery. The time point of the sham-operated fish was chosen based on the assumption that the surgical impact is the greatest at the beginning of the response. For every time point after cryoinjury and all control groups (sham and healthy), we obtained matching triplicates for mRNA and miRNA, whereas RNA was extracted and pooled from 3–6 whole hearts per sample. We excluded two mRNA and six miRNA samples that failed quality controls (see material and methods and Appendix A, respectively). Data was then pre-processed, normalized and corrected for unknown batch effects (see Materials and Methods and Appendix A).

Principal component analyses (PCA) of both the mRNAome and the miRNAome show consistent clustering over time, suggesting a well-organized and time-point specific regeneration process. For the mRNAome, early time points (1–30 dpi) of the cryoinjured fish separated from healthy, sham-operated and cryoinjured fish at late time points (≥45 dpi) according to the first principal component (Figure 2A, PC1 26%). A more detailed view suggests first an increase of the regeneration response from cryoinjured fish at 1 dpi to 4 dpi (path from left to right), with the latter clustering further away from healthy fish. Next, we see a gradually decline of the response from cryoinjured 4 dpi to 160 dpi and eventually healthy fish (path from right to left). Similar clustering can be observed for the miRNAome, where we see an increase of the response from cryoinjured fish at 1 dpi to 21 dpi (path from right to left), which is then followed by a decline back to 160 dpi (path from left to right). In contrast to the mRNAome, healthy fish further form a disparate cluster in the miRNAome (Figure 2B), which could indicate a permanent change in the miRNAome. Interestingly, sham-operated fish did not cluster with healthy fish for both the mRNAome and the miRNAome, indicating a large surgical influence of the transcriptome.

We hypothesize that sham-operated controls at 1 dpi are important initially, but become less suitable for late time points and healthy fish are preferred. This can be seen when comparing the transcriptomes of sham-operated fish to cryoinjured fish at 160 dpi. Up-regulated genes in sham-operated controls were enriched in response to wounding, regeneration and extracellular matrix (ECM) organization, suggesting a healthier state for the cryoinjured fish at 160 dpi and thereby indicate that the surgical impact became negligible (Appendix A). Consequently, a mix of sham-operated and healthy fish, with decreasing influence of the first, was needed as controls for the cryoinjury time-course experiment to successfully extract cardiac regenerative responses.

In this context, wound size during regeneration has been vastly described by an exponential decaying function [27], suggesting a similar behavior of the influence of the sham-operated fish as controls. Thus, we assigned time-dependent weights modeled by an exponential function, for sham-operated and healthy fish to describe the control group with respect to time (Materials and Methods; Figure 3A). Moreover, four groups of healthy fish with differing ages were included. This increases the variance of age-related genes in the control group and therefore lowers their chance of being differentially expressed when comparing to the cryoinjured fish. Having set the contributions of sham-operated and healthy fish as controls, differentially expressed genes (DEGs; FDR < 0.01 and |log2FC| > 1) were obtained for the mRNAome at each time point (Figure 3B) between the cryoinjured fish and the modeled control group. The largest peaks were observed at 4 and 7 dpi with 4266 and 3485 DEGs followed by a decrease to <1520 DEGs at 14–45 dpi and <1000 at 60–160 dpi. Interestingly, hierarchical clustering of log2-fold changes (cryoinjured vs. control) confirmed the three responses (Figure 3C); early (1–7 dpi), intermediate (14–45 dpi), and late (60–160 dpi). Using Gene Ontology (GO) gene sets, up-regulated DEGs in cryoinjured fish were highly enriched in multiple proliferation processes (*p* < 10^−9^) as well as in immune response and migration in the early response, whereas down-regulated DEGs showed enrichment in response to temperature stimulus, heart contraction and anion transport.

Proliferation processes were still up-regulated in the intermediate response (*p* < 0.01) but processes involved in ECM were more dominant. Ribosome biogenesis was severely down-regulated. At late time points we found increased enrichment in up-regulated genes involved in processes restoring heart functions, such as heart contraction, angiogenesis, and cell growth. On top of the aforementioned processes, enrichment analysis included more up- and downregulated processes in the early, intermediate and late phase of zebrafish heart regeneration that we would like to leave to the reader for further inspection (full lists see Appendix A).

Taken together, we established a linear model to identify specific cardiac regenerative changes between cryoinjured fish and time-dependent ratios of control samples, accounting for both age-related variations and the surgical impact. The biggest changes were observed for early time points with particular proliferation processes being activated.

### 2.2. Spatio-Temporal Organization during Cardiac Regeneration

Next, we sought to gain further insight into the time-course data to identify the major responses and its dynamics during heart regeneration and subsequently find the key miRNA regulators for those processes. We used soft-clustering of z-score transformed log2-fold expression changes of DEGs in at least one time point. Five clusters were identified showing distinct responses during cardiac regeneration in zebrafish (Figure 4A). Considering only genes with a cluster association of >70%, we assigned 4802 of 6955 genes uniquely to one of the clusters.

On the one hand, clusters 1, 2 and 4 show an increase in log2-fold changes peaking at 4 dpi, 4–7 dpi, and 21–30 dpi, respectively. The fastest response shows cluster 1, being back to baseline at 21 dpi and having slightly decreased levels at 160 dpi compared to the control group. While cluster 2 decreases back to baseline at 45 dpi, cluster 4 slowly increases and decreases while maintaining elevated expression at 160 dpi. For genes in cluster 1, we found significant enrichment for DNA repair, cilium organization and cell cycle mechanisms. Cluster 2 and 4 were both associated with immune response and chemotaxis, while cell cycle process, DNA replication, and regeneration were majorly related to cluster 2 and cell adhesion unique to cluster 4.

On the other hand, clusters 3 and 5 display an initial decrease, reaching its minimum at 4–7 dpi and then return to baseline at 21 dpi and 60 dpi, respectively. Cluster 5 remains at baseline, while cluster 3 continues to increase up to 160 dpi. Genes in cluster 3 were enriched for heart development/function, muscle cell development and actin filament processes, while cluster 5 showed an enrichment in RNA processing and translation (Figure 4B).

In order to get an idea of the primary location of these processes, the clusters were further linked to the specific transcriptomic profiles of the injury site, the border zone and the uninjured myocardium [15]. The different transcriptomic profiles were identified by Wu et al. using tomo-sequencing to whole hearts of zebrafish at 3 dpi. Briefly, RNA-seq expression data was obtained for various sections of the heart and clustering revealed the three profiles mentioned above. For every location profile, they ranked genes according to their absolute expression. We were able to identify the location of primary expression of our clusters through comparison of the ranks of the different profiles for each cluster. Genes in cluster 2, presenting mainly proliferation processes, were highest ranked in the injury area, meaning they showed the highest expression in the injury site. Cluster 3, mainly linked to heart function, was mostly associated with the uninjured myocardium, while cluster 1, 4 and 5 were similarly distributed among all the locations (Figure 4C).

Concluding, this shows the spatio-temporal organization of mRNAs during zebrafish heart regeneration, giving an in-depth view on dynamics, function and location of gene responses.

### 2.3. Identification of miRNA-mRNA Interactions during Cardiac Regeneration

MiRNAs have been shown to play an important role in cardiogenesis [19], in particular for cell cycle regulation [28], making them potential therapeutic targets.

To identify miRNA target interactions, we obtained theoretically predicted miRNA-mRNA interactions from TargetScanFish6.2 [29] with a context score < −0.2 (efficacy of binding site; the smaller the higher the confidence). However, many of the theoretically predicted sites are expected to be false positives [30]. Therefore, we calculated Pearson correlations between miRNA and mRNA expression of theoretical interaction partners. Then we kept only interactions that showed a significant negative correlation (FDR < 0.05 and ρ < −0.4), indicating a validated silencing complex in the transcriptome data.

We restricted our analysis to genes that could be assigned to a dynamic cluster with 70% certainty (n = 4802; Figure 4A) and miRNAs passing the filtering criteria in the preprocessing step (n = 285). In total 2660 interactions were predicted between 139 microRNAs and 1269 mRNAs (Appendix A). If miRNAs shared a significant number of target genes (hypergeometric test *p*-value < 0.01), they were combined into one group, which is often the case when they belong to the same miRNA family, e.g., miR-148 and miR-152, or being isoforms to each other. Figure 5 visualizes the top 10 miRNAs with the most mRNA interactions during zebrafish heart regeneration and functional enrichment was calculated for their target genes. Dynamics of miRNAs show an expected inverse response to the clusters of their target genes (Appendix A). 

There were two miRNAs that were initially up-regulated with predicted target genes associated to the down-regulated clusters 3 and 5 and enrichment in anion transport processes (miR-144; n = 95) as well as ubiquitin-protein activities (miR-142a; n = 61). In contrast, other top regulatory miRNAs were down-regulated during heart regeneration and mostly targeted genes from the up-regulated cluster 2 that were shown to be associated with proliferation processes at the injury site of the heart. Specifically, major contributors were miR-218ab (n = 88), the miR-148/152 family (n = 86) and miR-19abcd (n = 78) with genes enriched in DNA replication, regeneration processes, wound healing and regionalization (full list: Appendix A).

In addition, we found miR-133abc, miR-26ab, miR-29b, and miR-101ab that were also previously identified to play a major role in cardiac regeneration in zebrafish [23,24,25,31]. These miRNAs specifically targeted genes enriched in endocrine system development, cell migration, cell cycle processes, and blood vessel development.

Concluding, multiple miRNAs through up- and down-regulation were identified as key regulators during cardiac regeneration, with most target genes involved in cell cycle responses that were mainly expressed within the injury site. We identified previously described candidates and new targets with potential cross-species application.

### 2.4. Validation of Players Involved in Proliferation in Differentiating H9c2 Cells

Despite all top hub miRNAs being broadly conserved across species, there may be substantial differences in their target genes. In mammals, most mRNAs have been shown to be conserved targets of miRNAs but fish were excluded from the analysis because they lacked sufficient alignment in 3′UTR regions [22]. Therefore, we analyzed the dynamics of both mRNAs and miRNAs that are involved in zebrafish heart proliferation in a H9c2 cell line, that change from a proliferating to a differentiated phenotype, to gain insight in their behavior during proliferation processes in rat. It was shown in several studies that the usage of H9C2 can mimic the behavior of cultured primary cardiomyocytes [32,33].

H9c2 cells were derived from the lower half of a 13-day old embryonic rat heart (mostly ventricular) and myoblasts were selected by serial passaging. These myoblasts show a proliferating phenotype but when confluence increases (>90%) and differentiation supplements are added, myoblasts start to differentiate into non-proliferating cardiomyocytes-like cells. We measured mRNA and miRNA in triplicates of undifferentiated, proliferating myoblasts (70% confluence), the differentiation starting point (d0 = 90% confluence), two days (d2) after day zero (d0), and at five days (d5) to monitor dynamics with decreasing proliferation (Figure 6A). Thus, genes involved in proliferation processes are expected to become down-regulated over time, while their miRNA counterparts are up-regulated.

This can be confirmed by the fact that significantly down-regulated genes (FDR < 0.01 and log2FC < −1) in the differentiated rat CM-like cells (d5; non-proliferating phenotype) vs. undifferentiated myoblasts (undiff; proliferating phenotype) were significantly enriched in proliferation processes such as nuclear division, DNA replication, and meiotic cell cycle processes. Furthermore, among these we found a significant association (𝟀^2^
*p*-value < 10^−15^) to the genes in cluster 2 when mapped to zebrafish homologues (Figure 6B). Similarly, there exists an overall linkage (𝟀^2^
*p*-value < 10^−8^) between the target genes of the down-regulated miRNAs (Appendix A) during zebrafish heart regulation and the down-regulated genes in the differentiated rat CM-like cells, as well as an individual association (𝟀^2^
*p*-value < 0.05) for the target genes of miR-148/152, miR-19abcd, and miR-29b (Figure 6C).

Interestingly, we found only the miR-133 and miR-19 family as well as miR-26a expressed in the H9c2 cell line among the identified miRNAs from zebrafish heart regeneration (Appendix A), suggesting a potentially new set of miRNAs regulators involved in the H9c2 proliferation process. Hence, we assessed possible miRNA regulators in rat that influence the mRNAs involved in proliferation during zebrafish heart regeneration. We predicted miRNA-mRNA interactions for the genes that could be mapped to cluster 2 (n = 752) and calculated Pearson correlations between average expression levels of miRNA and mRNA at each time point of the H9c2 differentiation process. We restricted our analysis on theoretical predicted miRNA-mRNA interactions (TargetScanRat7.1) and kept interactions that showed a clear negative correlation (ρ < −0.4). Indeed, among the top five miRNAs with the most target genes, we identified four new players (miR-206, miR-128, miR-125a and miR-34c) and one common player, miR-133ab (Figure 6D), compared to the miRNAs involved in the zebrafish heart regeneration (Figure 5).

Taken together, we identified similar genes involved in proliferation processes of zebrafish heart regeneration and H9c2 differentiation, however, the important miRNA players were different with one exception: the miR-133 family. Further validation experiments in the zebrafish model are needed in order to identify and validate the true meaning of the individual miRNAs in the process of heart regeneration.

## 3. Discussion

An increased regenerative potential of the human heart following ischemic damage may reduce the occurrence of heart failure in patients. Here, we delineated the short- to long-term transcriptome response in zebrafish hearts after cryoinjury to gain insight into the mRNA dynamics and the driving miRNA regulators during zebrafish heart regeneration. To the best of our knowledge, this is the largest study of multi-omics data monitoring heart regeneration up to 160 dpi in zebrafish. While cryoinjury has been shown to be a good model to mimic myocardial infarction [6,8] the impact of the preceding surgery clearly affects the transcriptome and has to be taken into consideration, e.g., with the use of sham-operated controls. In addition, transcriptome changes with increasing age need to be adjusted using age-matched controls. Hence, age-matched and sham-operated controls are recommended to best study the regeneration related response in zebrafish.

However, preparing sham-operated and age-matched controls for every time point increases the operating time and expenses notably. While other studies used either sham-operated controls which recovered at one time point [12,16] or healthy controls [34], we used a control group with time-dependent weights of sham-operated fish (recovered at 1 dpi) and unoperated healthy fish covering all ages to extract particular heart regeneration changes induced by cryoinjury. Yet, this approach underlies the assumption of an exponential decay of the surgery influence and only sham-operated controls at every time point can exactly mimic the surgical impact in the transcriptomic changes. Sham-operated controls may be particularly important for studying the changes of the miRNAome of cryoinjured fish at late time points because even at 160 dpi, samples were still separated from healthy controls (Figure 2B). Alternatively, this might suggest a possible permanent change of the miRNAome after heart injury.

Using a mixed-control group, differential expression analysis showed the largest transcriptome changes at early time points (1–7 dpi) suggesting a major and fast response to cryoinjury. The most prominent pathways correspond to cell cycle and DNA replication processes, which is in accordance to previous studies [4,12,16] and the fact that regeneration occurs mainly by proliferating CM rather than differentiation processes [10,11]. During the intermediate response (14–45 dpi) cell cycle processes are still up-regulated, but ECM organization processes become more prominent, indicating a still ongoing remodeling of the scar tissue. At late time points (60–160 dpi) we observed no significant cell cycle alterations, suggesting a completion of the heart regeneration and tissue remodeling up to 60 dpi after cryoinjury. Yet, full functional recovery has not been accomplished, with heart function restoring processes still being up-regulated at 60–160 dpi, such as cardiovascular development, heart contraction and growth.

These results were further corroborated by the delineation of the transcriptome response using soft-clustering techniques. Here, proliferation regulators showed a fast increase up to 4–7 dpi and a slow decrease back to baseline until 45 dpi, which confirms previously observed phenotypical data of scar volume [7,8]. Once, the heart tissue is reestablished, processes restoring the heart function are activated. This can be seen for clusters 3 and 4, which did not return to baseline levels at 160 dpi and were being enriched for heart development and adhesion processes. Therefore, the transcriptome response of heart regeneration after cryoinjury is rather a long process up to 160 dpi [6], which is further in agreement with phenotypic observations of Hein et al. who still found limited radial wall displacement and the presence of scar tissue at 180 dpi [8].

Using spatially resolved transcriptome data [15], we assigned the dynamic clusters with their primary location sites in the injured heart. Genes involved in proliferation processes were highest expressed within the injury site of the heart, evincing the injured tissue to be the major source of proliferation, possibly through invading CM from the uninjured myocardium [13].

Since miRNAs are suggested to play a major role in regulation of heart regeneration [19], we investigated both the mRNAome and the miRNAome. By correlating mRNAs with miRNA dynamics, we identified miRNA target genes with visible repression (anti-correlation) in expression and therefore diminished false-positive binding sites that are often present in miRNA target-interaction data bases [21]. Among the top 10 miRNAs regulating the most genes, we found previously described and potential novel candidates.

We identified miR-144 to be up-regulated in the initial response (1–21 dpi; Appendix A) and targeting mostly genes in clusters 3 and 5 that were initially down-regulated. Interestingly, miR-144 was shown to play an important role in hematopoiesis and vascular development in zebrafish through inhibition of meis1 [35], but in our data both miR-144 and meis1 were up-regulated up to 21 dpi, suggesting a different mechanism. However, we could confirm enrichment of GO:0048534 hematopoietic or lymphoid organ development for down-regulated genes in the early, late and intermediate response, what matches to the dynamic of miR-144 and its role in hematopoiesis. While it could be hypothesized that these miRNAs hinder the regeneration process, we believe that the down-regulation of genes in clusters 3 and 5 belongs to a well-orchestrated regeneration process and is initially needed when proliferation is most important.

On the other hand, miR-218b was initially down-regulated and its target genes (up-regulated) were associated with DNA replication (Appendix A). In cancer miR-218 acts as a tumor suppressor by inhibiting proliferation and migration in glioma cells [36], bladder cancer [37] and non-small cell lung cancer [38], indicating similar mechanisms regarding proliferation.

Moreover, the down-regulation of miR-148/152 family could be linked to the up-regulation of genes mostly involved in proliferation processes (cluster 2), suggesting an inhibitory role of proliferation during cardiac regeneration in zebrafish. Also, in carcinogenesis miR-148 was found to inhibit proliferation by targeting pro-proliferative genes and thereby acting as a tumor suppressor [39], pointing to a similar role and its multifunctional potential. It was also linked to suppress migration and invasion in tumor cells [39], which again plays an important part in heart regeneration [13]. At 160 dpi, miR-148/152 were up-regulated, suggesting a possible stopping mechanism of proliferation processes.

Furthermore, we identified down-regulation of the miR-19 family as key regulation of increased proliferation (cluster 2), with miR-19b being previously observed as down-regulated during zebrafish heart regeneration [24].

Among the known miRNA regulators of zebrafish heart regeneration, we confirmed the important role of miR-133 family, whose depletion has been shown to enhance CM proliferation [23]. MiR-26a has been shown to target cell cycle activators and inhibition stimulates cardiomyocyte proliferation in post-natal mouse hearts [25] underlining its importance in both zebrafish and mice heart regeneration.

For miR-101a, we could corroborate similar expression dynamics in injured zebrafish hearts as previously reported [24]. While miR-101a is initially (1–3 dpi) down-regulated post-amputation and shown to enhance CM proliferation [24], it remains down-regulated up to 120 dpi post cryoinjury. At later time points post-amputation (7–14 dpi), miR-101 becomes up-regulated compared to uninjured hearts, which could be associated with scar tissue removal [24]. In cryoinjured fish, however, up-regulation does not occur until 160 dpi, suggesting a similar yet much slower regenerative response compared to resection experiments (Appendix A).

Another known candidate is miR-29b that inhibits genes involved in ECM, confirming results in mice, where down-regulation of the miR-29 family was shown to regulate fibrosis after myocardial infarction [31].

Also, miR-142a has been found to play an important role in vascular integrity and developmental angiogenesis, with overexpression leading to a loss of these functions [40]. Interestingly, miR-142a is up-regulated during heart regeneration, suggesting miR-142 is an antagonist to heart regeneration. However, loss of these processes might also be necessary for the beginning of the response.

Last but not least, we identified miR-16c as novel player of heart regeneration in zebrafish with an association to cell cycle processes.

To research the cross-species potential of the identified candidates in proliferation processes of zebrafish heart regeneration, we compared our results to a differentiating H9c2 cell line that loses its proliferative capability with ongoing differentiation [33]. Interestingly, we found a significant association between the genes of cluster 2 in zebrafish and the down-regulated genes in differentiated rat CM-like cells, with both of them involved in proliferation processes. Furthermore, the target genes of the miRNAs involved in zebrafish heart regeneration were related to the down-regulated genes in the non-proliferative phenotype of the differentiated CM-like cells, indicating similar mechanisms regarding proliferation. However, the miRNAs associated with proliferation in the H9c2 cell line included only the miR-133 family as common player between zebrafish and rat, strengthening its important role in proliferation for both species and both experimental designs. Moreover, the analysis included miR-125a, which was found to inhibit proliferation in C2C12 myoblasts by targeting E2F3 [41], an interaction we could confirm in the H9c2 cell line (Figure 6D) and miR-128, which loss was identified to promote CM proliferation and heart regeneration in postnatal mice [42]. The discrepancies between the two studied organisms may be explained by (i) the experimental setups: in vivo vs. in vitro. Most miRNAs that played an important role in the in vivo setting of zebrafish were not expressed in the in vitro setting of the H9c2 cell line. In contrast, all miRNAs were expressed in an in vivo mouse model [25]. (ii) The cell composition: whole hearts vs. myoblasts that differentiate into CM-like cells, (iii) the differences in miRNA binding sites between fish and rat [22] and (iv) the proliferation trigger/stop: cryoinjury vs. differentiation. All of these can result in different miRNA expression, impeding comparative analysis between the two systems. Moreover, in order to overcome these discrepancies, we will carry out further validation experiments of predicted candidate miRNAs in zebrafish, e.g., using morpholino oligos to block miRNA, to understand the actual influence of the predicted miRNAs in cardiac regeneration.

## 4. Materials and Methods

### 4.1. Zebrafish Care and Cryoinjury

Zebrafish care, breeding and cryoinjury experiments on 3–6 months old fish were performed as described previously [8,43]. All fish had the same genetic background and experiments were conducted with approval of the local Animal Care Committee and according to institutional guidelines.

### 4.2. Cell Culture

H9c2 cells (ATCC #CRL-1446) were cultured in low glucose DMEM (Sigma #D5796) supplemented with 10% FCS (Biochrome #S0615), 50 U/mL penicillin and 50 µg/mL streptomycin (Thermo Fisher Scientific #15070-063). For differentiation, cells were allowed to reach >90% confluence and medium was switched to 1% FCS with additional 10 nM retinoic acid (Sigma #R2625). Medium was renewed every second day during differentiation.

### 4.3. RNA-Seq and Preprocessing

Hearts were extracted from healthy, sham-operated and cryoinjured zebrafish at 1–160 dpi. For each time point triplicates were obtained with each sample containing RNA from 3–5 hearts. For both zebrafish hearts and H9c2 cells, RNA was extracted using RNeasy Mini Kit (Qiagen #74104). 200 and 500 nanograms were used for library preparation with TruSeq Stranded Total RNA Library Prep (Illumina #20020597) for mRNA for zebrafish hearts and H9c2 cells respectively. The miRNAs were extracted using miRNeasy Mini Kit (Qiagen #217004) and QIAshredder (Qiagen #79654) and used for library preparation with the TruSeq Small RNA Library Prep (Illumina #20005613) for zebrafish hearts and H9c2 cells. All kits were utilized according to manufacturer’s instructions and samples were send to sequencing at the NGS unit of Prof. Dr. Benjamin Meder, University hospital Heidelberg.

Paired-end reads were obtained for mRNA and single-end reads for miRNA samples.

Resulting raw FASTQC files were trimmed for bad quality reads and adapter sequences using trimmomatic [44]. The aligner STAR [45] was used to align reads to the Ensembl GRCz10 Genome and obtain a count matrix for Ensembl IDs and samples. Raw files for the H9c2 cell line were trimmed and then aligned to the Ensembl Rnor6.0 genome. Outliers were detected by visual inspection of quality control plots and removed from further analysis when they failed 4 out of 6 quality controls (Appendix A). Subsequently, Ensembl identifiers (IDs) were matched to Entrez IDs. In case multiple Ensembl IDs matched to one Entrez ID, the one with the largest inter-quartile-range across samples was kept. Data was filtered for low count reads (> one count per million in at least three samples), TMM normalized, corrected for batch effects using surrogate variables [46] (Appendix A) and tagwise dispersions were calculated for statistical analysis using the R/Bioconductor package edgeR [47]. Raw data is accessible at https://www.ncbi.nlm.nih.gov/sra/PRJNA509429.

### 4.4. Mix-Control Modeling and Statistical Analysis

To extract heart regeneration specific changes within the transcriptome, we compared the time-course of cryoinjured zebrafish to a control group consisting of time-dependent ratios of healthy and sham-operated zebrafish. Assuming exponential recovery from surgery stress [27], we modeled the control group as follows:(1)$$\mathrm{control}\left(t\right)={\;w}_{\mathrm{sham}}\left(t\right)\cdot \mathrm{Sham}+w_{\mathrm{healthy}}\left(t\right)\cdot \mathrm{Healthy},$$
with $w_{\mathrm{sham}}\left(t\right)=e^{-\beta \left(t-1\right)}$ and $w_{\mathrm{healthy}}\left(t\right)=1-w_{\mathrm{sham}}\left(t\right),$ assuming $w_{\mathrm{sham}}\left(t=1\mathrm{dpi}\right)=1$ and $w_{\mathrm{sham}}\left(t=160\;\mathrm{dpi}\right)=0.05$.

Differentially expressed genes were defined as FDR < 0.01 and absolute log2-fold change (log2FC) > 1 and calculated with the R/Bioconductor package edgeR [47]. Gene set enrichment of Gene Ontology (GO) terms was calculated using fisher tests for a specified gene set compared to all background genes. Significance was defined as Fisher’s Test *p*-value < 0.05 and further a minimum overlap of four genes between the GO term and the gene set of interest.

### 4.5. Spatio-Temporal Organization

Gene dynamics were soft-clustered into five groups by clustering z-score transformed log2FC between cryoinjured fish and their respective control group, using the R/Bioconductor package mfuzz [48]. For further analysis, we considered genes (n = 4802) with a cluster association of > 70%. For each cluster, gene set enrichment was calculated using GO terms. Clusters and GO terms were connected in a graphical network with edges drawn if either GO terms were significantly associated to a cluster (*p*-value < 0.05) or if GO terms share at least 40% of their genes. Wu et al. [15] ranked genes according to their absolute expression in the injury site, border zone and uninjured myocardium of regenerating zebrafish hearts. For each cluster i, we matched the genes to the ranks of the location j and calculated for each cluster a location weight according to:(2)$$w_{\mathrm{ij}}=\frac{\max\left({\mathrm{ranks}}_{\mathrm{ij}}\right)}{\max\left(\mathrm{Top}\;20{\%\;\mathrm{of}\;\mathrm{ranks}}_{\mathrm{ij}}\right)}.$$

Therefore, if many genes in one cluster are ranked as high for a specific location, it will be assigned a large weight and vice versa. Subsequently, clusters were arranged dynamically using the graph network visualization program Gephi with the layout algorithm ForceAtlas2, which takes into account edge weights.

### 4.6. miRNA-mRNA Predictions

First, theoretically predicted miRNA-mRNA interactions were obtained from TargetScan with a context score < −0.2. Secondly, for each interaction Pearson correlation coefficients were calculated across expression from all samples with both miRNA and mRNA data and the interaction was considered valid during zebrafish heart regeneration for ρ < −0.4 and FDR < 0.05. Because we did not have matching miRNA and mRNA data for the H9c2 cell line, first, expression levels of time points (undiff, d0, d2, d5) were averaged across triplicates, and second Pearson correlations were calculated between mRNAs and miRNAs expression for the different time points and considered valid for ρ < −0.4. We restricted our analysis on mRNAs that could be mapped to cluster 2 and miRNAs that were significantly up-regulated (FDR < 0.01 and logFC > 1) in d5 vs. undiff. Therefore, only miRNA-mRNA interactions were considered that most likely play a role in the decrease of proliferation during H9c2 differentiation. 

### 4.7. Homologue Mapping between Zebrafish and Rat

Gene homologues between zebrafish and rat were found by mapping Ensembl IDs between species using the Bioconductor package biomaRt [49].

## 5. Conclusions

We present a detailed and dynamic picture of both the mRNA and miRNA changes during the long process of zebrafish heart regeneration and identified important miRNA regulators of cardiac proliferation following ischemic damage. In a cross-species effort, we compared our findings to the decreasing proliferative phenotype in a differentiating H9c2 cell line, which suggested a similar role of mRNAs during proliferation. However, only the miR-133 family shows consistencies between zebrafish and rat on the miRNA level. Additional work is needed and planned to validate the dynamic and spatial resolution of mRNAs and if knock-down/out experiments of the identified miRNAs have an impact on heart regeneration in vivo in cardiomyocytes and in mouse models. Furthermore, we want to analyze whether we can push cardiomyocytes to re-enter the cell cycle in a non-regenerative in vivo setting and therefore test their therapeutic potential.

## Figures and Tables

**Figure 1 biomolecules-09-00011-f001:**
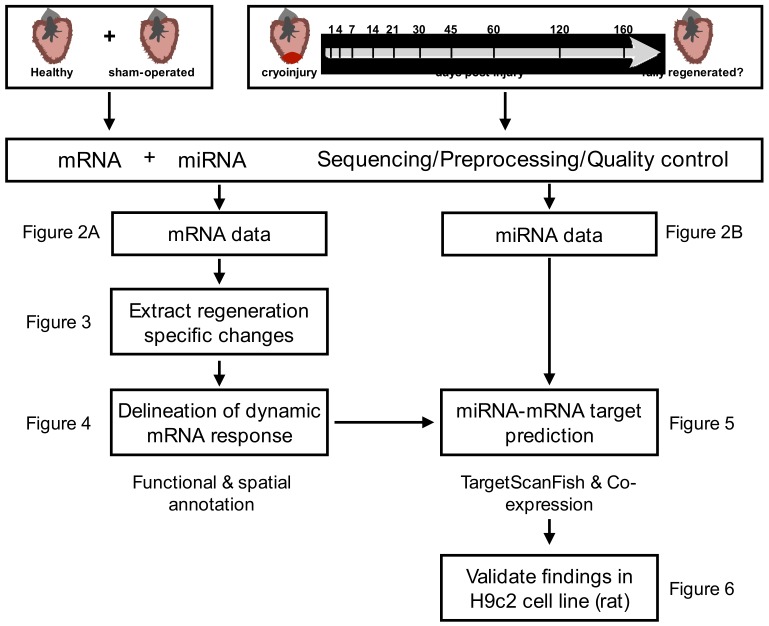
Schematic workflow of the study design and the analysis steps.

**Figure 2 biomolecules-09-00011-f002:**
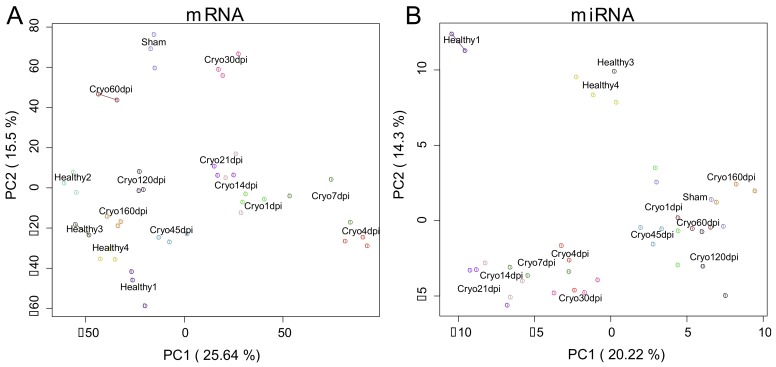
Principal component plots of log2-expression (counts per million) of (**A**) mRNA samples and (**B**) miRNA samples. For each sample RNA was extracted from 3–5 zebrafish hearts and both mRNA and miRNA were sequenced.

**Figure 3 biomolecules-09-00011-f003:**
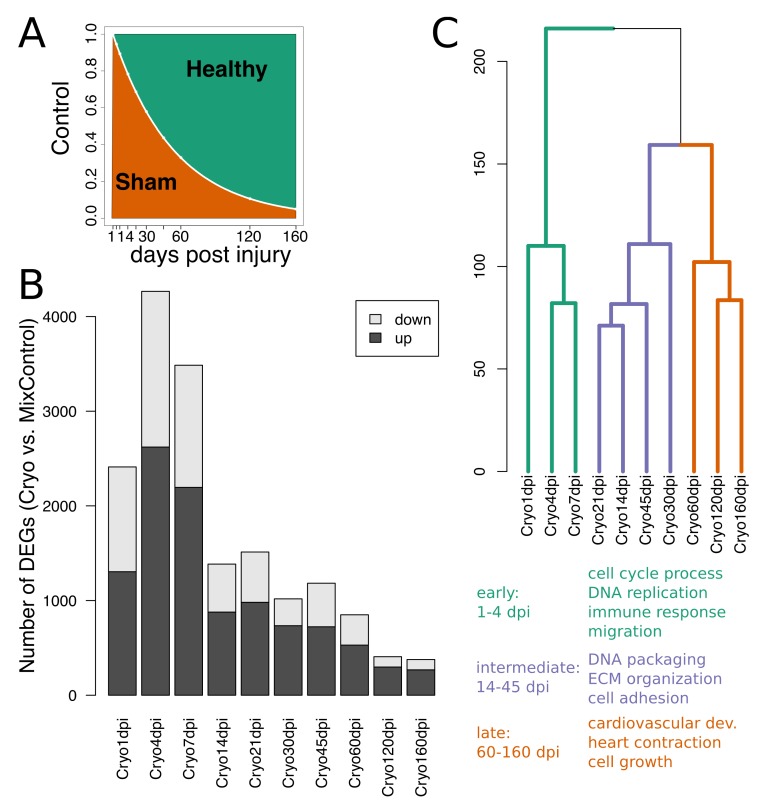
(**A**) Ratios of healthy and sham-operated fish in control group. (**B**) Number of up- and down-regulated genes over time. (**C**) Hierarchical clustering of log2FC between cryoinjured fish and control group. Significant gene sets for DEGs (FDR < 0.01 and |log2FC| > 1) in early (1–7 dpi), intermediate (21–45 dpi) and late (60–160 dpi) cryoinjured fish.

**Figure 4 biomolecules-09-00011-f004:**
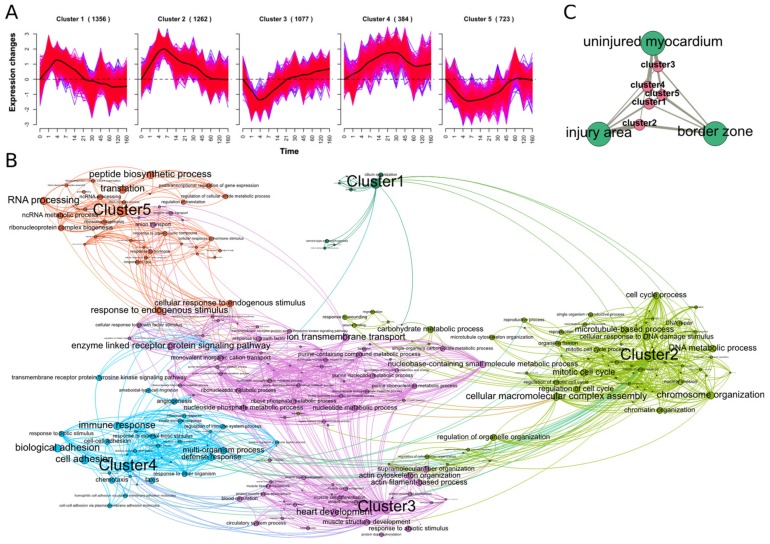
(**A**) Soft-clustering of log2FC over time. Black lines visualize the average dynamics of the clusters. (**B**) Gene set enrichment analysis (Fisher’s Test) of genes from every cluster. Gene sets are connected by an edge if they share 30% of genes. (**C**) Association of clusters to the different profiles after cryoinjury identified by Wu et al. [15].

**Figure 5 biomolecules-09-00011-f005:**
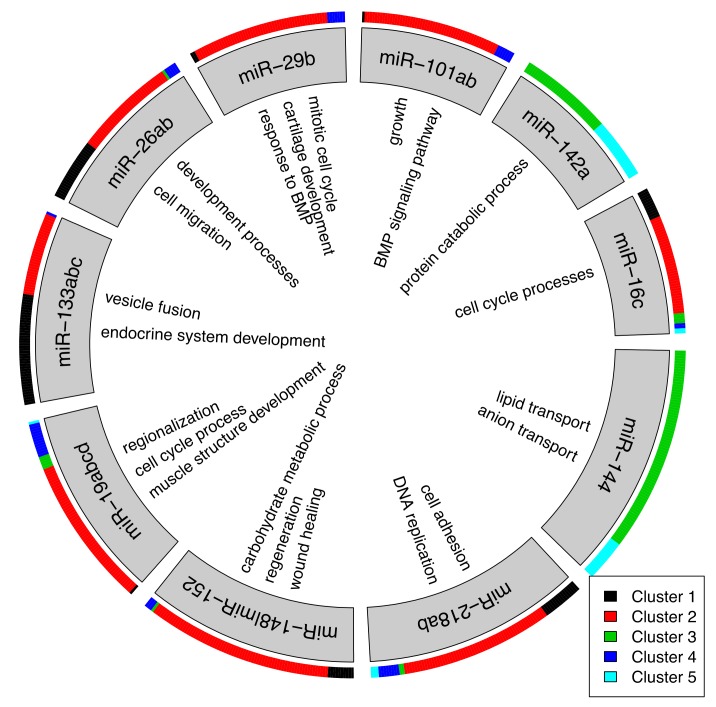
Top 10 miRNAs with the most predicted target genes from clusters 1–5 (Figure 4A). miRNAs were combined if they shared a significant overlap of target genes, e.g., miR-148 and miR-152 (hypergeometric Test *p*-value < 0.05). A selection of Gene Ontology (GO) terms are shown in inner circle that were enriched for the target genes. A full list of all mRNA-miRNA interactions can be found in Appendix A.

**Figure 6 biomolecules-09-00011-f006:**
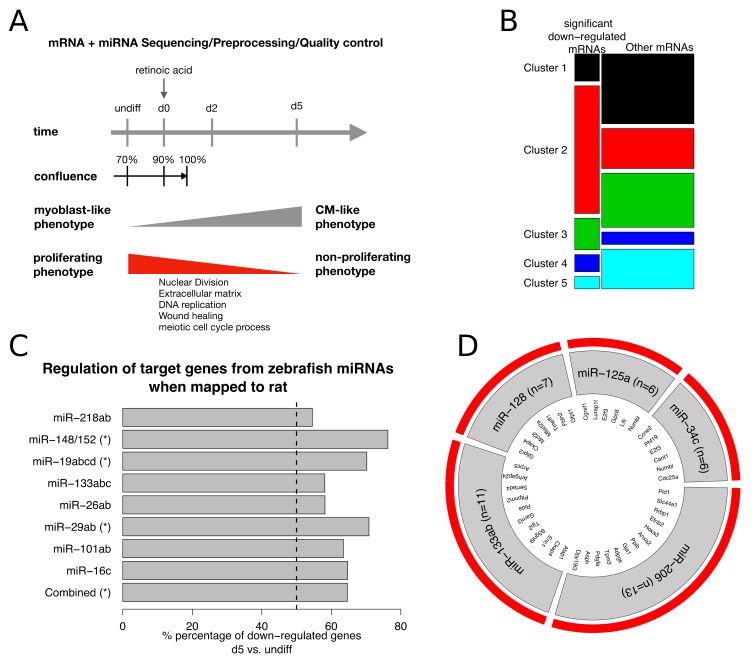
(**A**) Experimental design of the H9c2 cell line. Both mRNA and miRNA were sequenced at 4 different time points that display a differentiation into a CM-like phenotype and a reduction of the proliferative capability. (**B**) Proportions of the significant down-regulated genes (d5 vs. undiff; FDR < 0.01 and log2FC < −1) and the remaining mRNAs in the H9c2 cells when mapped to zebrafish homologues and their corresponding dynamic cluster in zebrafish heart regeneration. Cluster 2 is significantly overrepresented in the down-regulated genes (𝝌^2^
*p*-value < 10^−15^). (**C**) Percentage of down-regulated genes in differentiated CM-like cells (rat) when mapped to the target genes of miRNAs that were identified in zebrafish heart regulation. (*) indicates a significant proportion of down-regulated genes (𝝌^2^
*p*-value < 0.05). (**D**) Top 5 miRNAs with the most predicted target genes in differentiating H9c2 cells. mRNAs were restricted to rat homologues from the dynamic cluster 2 in zebrafish heart regulation and miRNAs were significantly up-regulated at d5 (FDR < 0.01).

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
