# Peer review of "Delineating the Dynamic Transcriptome Response of mRNA and microRNA during Zebrafish Heart Regeneration"

_biomolecules, 2018, doi:10.3390/biom9010011_

Round 1
Reviewer 1 Report
The manuscript entitled “Delineating the dynamic transcriptome response of mRNA and microRNA during zebrafish heart regeneration” by Klett et al. describes the regeneration specific transcriptional responses in the adult zebrafish heart after myocardial cryoinjury. Additionally, the authors correlated the obtained miRNA and mRNA expression profiles to predicted potential regeneration-specific miRNA-mRNA interactions. Finally, they validated their findings from the adult regenerating zebrafish heart in a cross-species approach by investigating a mammalian cardiomyocyte line (H9c2 cells).
The paper is well written, the topic interesting and the experimental setup appropriate to gather new insights into the transcriptional responses in the adult regenerating zebrafish heart. The presented work contains interesting findings regarding the regeneration-specific transcriptional responses in the zebrafish heart.
Nevertheless, prior publication, the manuscript should be revised.
Specific points:
1. Although broaching the issue of the used controls in the discussion section, the controls used in this study are still somewhat disputable (at least for this reviewer) and their validity should be explained in more detail.
- Why the authors didn´t use age-matched sham-operated siblings as specific controls for their cryoinjured fish at the 10 different time-points? This was just a question of costs? This point is even more relevant since the authors describe a surgery-associated impact on both, mRNA and miRNA profiles. The surgery-associated impact on transcription would be interesting and important also at later stages and not only at 1 dpi to really extract the pure regeneration-linked transcriptional changes. Please extent your statements.
- Are all used fish from the same genetic background? This point might be important to exclude potential background-specific genetic/transcriptional/regenerative differences and heterogeneity. Please comment on this.
2. Did the authors investigate and validate up- or downregulation of specific transcripts (e.g. Cluster-defining transcripts) by qRT-PCR or in situ hybridization on injured hearts (heart sections)? This would (1) prove their findings and (2) demonstrate spatial distribution in the regenerating zebrafish heart. This would be even more interesting since it was surprising (at least for this reviewer) that e.g. Cluster 2 genes (mainly linked to proliferation) appear to locate predominately to the injured tissue and not the highly proliferative wound border zone in the regenerating adult zebrafish heart.
Minor points:
- Abstract: The state-of-the-art cardiac injury techniques in the adult zebrafish heart such as cryoinjury, cardiomyocyte ablation or partial ventricular resection do not represent ischemic cardiac damage. The authors should rephrase this sentence.
- In some figures (e.g. Fig.1 (fully regenerated?) or Fig. 2 (Cryo 1dpi)), text was covered by the picture.
Reviewer 2 Report
Summary:
The authors have measured mRNA and miRNA levels of cryoinjured, sham-operated and healthy hearts of zebrafish over a time course for 160 days to observe changes due to heart regeneration. PCA analysis revealed a separation of early and late processes during regeneration indicating a time-specific regulation of the process.
Clustering of the mRNA data further refined this observation; three separate stages of regeneration could be observed and different molecular processes could be attributed to these stages by gene enrichment analysis.
To elucidate the role of miRNA regulation of mRNA, soft-clustering of the miRNA data was performed. Five clusters were found and for each their temporal and spacial dynamic was analysed, revealing different groups of molecular processes which are regulated by said miRNAs.
In a last step, the ten miRNAs which had the broadest impact on the mRNAome were analysed regarding the cluster they belonged to and the genes they impacted most.
To see if these miRNAs were also relevant in the processes of cardiac regeneration in other species, similar experiments were performed on a proliferating and differentiating rat myocardial cell line.
Broad comments:
The work the authors present is very interesting, however, I do have quite a few questions regarding experimental setup and design decisions.
One of my biggest concerns were the PCA analyses (detailed in Figure 2). While I understand the conclusions the authors draw from the PCA analyses, I cannot fully agree with these conclusions.
1. Regarding the PCA analysis of the mRNAome: To me the separation into two stages seems a little artificial and not as clear as is suggested by the text. On the other hand, with the exception of 1dpi there is kind of a path from right to left following from injury to healthy which seems more like a gradual development from injury to healthy. This should be discussed in the manuscript.
2. Regarding the PCA analysis of the miRNAome: All of the injury samples are clearly separable from the healthy ones. This makes it important to follow the sham operation until the end to see if this is the result of cardiac ischemia or of the operation (e.g. if the operation induces a permanent change in miRNAome). This should be discussed in the manuscript as well.
3. Regarding the PCA analysis in general: Other issues I would like to see discussed in the manuscript:
- What does the anomalous behavior of the 1dpi samples mean?
- The sham samples display different behavior in mRNA and miRNA. This is contrary to your opinion that the sham samples become less important later on. Was there a PCA analysis with all of the sham samples without wheights?
- In Figure 2 for each mRNA and miRNA only a small group of sham samples is displayed. Are the profiles of the sham samples so similar that they all form one small cluster or did you only include the sham samples of one point in time in the figures?
4. The clustering of the mRNA data was very interesting to me (detailed in Figure 3). I only have two minor comments:
- In the "Results" section it is never mentioned which database you use for gene enrichment.
- The processes you describe, e.g. immune response and cell proliferation in the early phase, etc., are well known in zebrafish heart regeneration. The authors need to put more emphasis on the novelty and special features of their work.
5. Next, the authors performed a clustering of miRNA (Figures 4 and 5).
- Since miRNAs usually inhibit their targets this would mean that some of the miRNAs actively hinder the regeneration process. This would also explain why the genes in clusters 3 and 5 are decreased early in the regeneration process. You should definitively point this out more clearly.
- However, most of these broadly targeting miRNAs target genes of cluster 2 which are notably not downregulated in the beginning of the regeneration process. Why could that be? Please discuss this as well.
6. The spatial analysis, on the other hand, seemed out of place to me. Since the whole hearts of the zebrafish were taken and analysed this association seems unfitting regarding the data. The results are as expected. I agree that such an analysis may be interesting, however, I find the experimental setup in this manuscript is not suitable for the application that Wu et al describe. To make this experiment fit into the context of the experiment of Wu et al. the authors would need to adress these issues in more detail.
7. My other concern are the validation experiment in the rat myocardial cell line (detailed in Figure 6). The experimental setup as it is described rather resembles normal development and not necessarily regeneration of myocardial cells. Especially since adult zebrafish are used in the original experiments adult rats or mice would be better suited to test interspecies comparability. As a result of this mismatch between original experimental setup and the setup of the validation experiments the comparability between the two falls short. I am not sure how this qualifies as a validation experiment, and would even go so far as to say that the validation experiments undermine the hypothesis of the authors. These experiments need to be repeated with either a cell line which actually expresses the miRNAs found in the zebrafish experiments, or a comparable animal model e.g. the existing mouse model mentioned later in the manuscript (line 369) which even expresses all miRNAs found in the zebrafish analyses (contrary to the H9c2 cell line).
8. While it is nice to have "Results" and "Discussion" as separate sections it is easier to emphasise the significance of experiments if some of the meaning and impact of the results is discussed immediately after describing the results.
9. Something I found expecially interesting was the short descriptions of each of the miRNAs found in the analyses. The miRNAs 142a and 16c were not mentioned in the "Discussion", though. Does this mean there is no knowledge yet about these miRNAs? If yes then this is a discovery made by the authors and should be mentioned.
10. Finally, while the authors name all the issues which could be responsible for the discrepancies between the zebrafish and the rat cell line experiments (lines 393-398), none of them were discussed critically. A more detailed discussion of this and maybe some ideas for future experiments would improve the manuscript.
Specific comments:
- Lines 140, 275 and 277: probably a formatting mistake, should say p < 10^-9, p-value < 10^-15 and p-value < 10^-8 respectively?
- In Figure 2 the labeling is partially unclear and covers the graphs in places. This needs to be corrected. In Figure 2B I also cannot find the points for Healthy 2.
- Figure 3C: The term from the enrichment analysis should be integrated into the figure differently. In C you should also mention the grouping into early, intermediate and late response to cryoinjury.
- Lines 155-165: You mention age-related variations. However, they do not seem to influence the analyses nor does it seem as if it was tested for age-related differences. How do age-related differences influence your linear model of cardiac regeneration?
- line 203: If you expect many false positives with a context score of <-0.2, why not choose a lower threshold? Why did you choose this threshold?
- Figure 5: The terms on the right side (within the circle) should not be upside down, rather turn them around for better readability.
Reviewer 3 Report
The article by Klett et al. profiled the mRNA and microRNA transcriptome during zebrafish heart regeneration by RNA sequencing. Overall speaking, the manuscript studied an interesting and important process, the heart regeneration. The study was well-designed, and all bioinformatic and statistical analyses seem to be performed appropriately. However, there are some concerns.
1. There could be many better options to validate the miRNA candidates identified in fish regeneration than using H9c2 cell differentiation. The most obvious one, as the authors also point out in the manuscript, will be to investigate the impact of knocking-down/overexpressing the identified miRNA on zebrafish/adult mouse heart regeneration. Knock-down/overexpression could be done through injection of morpholino/miRNA mimic in fish. Depending on the exact role of the identified miRNA, we can expect either a faster regeneration or no regeneration in fish or increased regeneration in mouse. Adding one of these experiments will significantly strengthen the study and is highly recommended.
2. There does not seem to be description of data availability. Deposition of the original data to a public database like GEO is generally required these days. Please upload and add that info in the manuscript.
3. Some minor typos need to be fixed:
a. Line 275 and 276, “10-15” and “10-8”.
b. Line 439, Rnor6.0 is rat not fish. Please revise.
c. Figure 1 and 2, some texts were blocked. Please revise.
Round 2
Reviewer 3 Report
The manuscript is good to go.